# CFTR Inhibitors Display In Vitro Antiviral Activity against SARS-CoV-2

**DOI:** 10.3390/cells12050776

**Published:** 2023-02-28

**Authors:** Anna Lagni, Virginia Lotti, Erica Diani, Giada Rossini, Ercole Concia, Claudio Sorio, Davide Gibellini

**Affiliations:** 1Microbiology Section, Department of Diagnostic and Public Health, University of Verona, 37134 Verona, Italy; 2Microbiology Unit, IRCCS Azienda Ospedaliero-Universitaria di Bologna, 40138 Bologna, Italy; 3Department of Diagnostic and Public Health, University of Verona, 37134 Verona, Italy; 4General Pathology Section, Department of Medicine, University of Verona, 37134 Verona, Italy

**Keywords:** SARS-CoV-2, cystic fibrosis, CFTR, CFTR inhibitors, antiviral, IOWH-032, PPQ-102

## Abstract

Several reports have indicated that SARS-CoV-2 infection displays unexpected mild clinical manifestations in people with cystic fibrosis (pwCF), suggesting that CFTR expression and function may be involved in the SARS-CoV-2 life cycle. To evaluate the possible association of CFTR activity with SARS-CoV-2 replication, we tested the antiviral activity of two well-known CFTR inhibitors (IOWH-032 and PPQ-102) in wild type (WT)-CFTR bronchial cells. SARS-CoV-2 replication was inhibited by IOWH-032 treatment, with an IC_50_ of 4.52 μM, and by PPQ-102, with an IC_50_ of 15.92 μM. We confirmed this antiviral effect on primary cells (MucilAir^TM^ wt-CFTR) using 10 μM IOWH-032. According to our results, CFTR inhibition can effectively tackle SARS-CoV-2 infection, suggesting that CFTR expression and function might play an important role in SARS-CoV-2 replication, revealing new perspectives on the mechanisms governing SARS-CoV-2 infection in both normal and CF individuals, as well as leading to potential novel treatments.

## 1. Introduction

SARS-CoV-2 is an enveloped, positive-sense, single-stranded RNA virus belonging to the genus *Betacoronavirus*, and it is the etiological agent of coronavirus disease 2019 (COVID-19). This virus first appeared in Wuhan, China, in December 2019 and, despite containment attempts, expanded rapidly worldwide, creating an urgent need to identify effective antiviral drugs [1].

Intense fatigue, headaches, dyspnea, myalgia, and gastrointestinal symptoms such as vomiting, stomach pain, loss of appetite, and diarrhea are all common symptoms of COVID-19. Moreover, certain other symptoms, including hyposmia, anosmia, ageusia, maculopapular rash, and urticarial lesions, have also been reported. Cases with severe clinical evolution are related to cytokine storm syndrome, a derangement of the regulation of certain cytokines such as IFN, MCP1, IP-10, TNF-α, and IL-10. This dysregulation causes local tissue damage and systemic non-protective inflammation, which may lead to sepsis and lung injury, often involving acute respiratory distress syndrome (ARDS), pneumonitis, respiratory failure, sepsis shock, organ failure, and possibly death [2,3].

Several molecules were considered in in vitro studies to determine their efficacy and safety as potential agents for COVID-19 treatment. Corticosteroids, antivirals, interferons, and monoclonal antibodies directed against immune system proteins, such as interleukin-6 (IL-6), or against certain SARS-CoV-2-specific targets were investigated in order to tackle either the viral replication cycle or the inflammation related to infection [4,5,6].

Currently, a few treatments are recommended against SARS-CoV-2; specifically, in the case of non-severe disease but a high risk of hospital transmission, the WHO recommends the use of Nirmatrelvir (Paxlovid), an active protease inhibitor [7] associated with Ritonavir. In addition, the ribonucleoside analog Molnupinavir and the viral polymerase inhibitor Remdesivir are effective in the inhibition of SARS-CoV-2 replication, with a favorable safety and tolerability profile [8,9], and both molecules are conditionally recommended by the WHO in cases of non-severe and severe COVID-19 [10]. Although the onset of new variants has partially counteracted the antiviral activity of several monoclonal antibodies towards S viral protein, they are still consistently used for SARS-CoV-2 infection treatment.

Recent reports have shown that SARS-CoV-2 infection does not result in worse outcomes in people with cystic fibrosis (pwCF); in a multinational cohort study of 40 CF patients, the incidence of COVID-19 in pwCF was 0.07%, compared to 0.15% in the general population [11]. Moreover, an international cohort analysis of 181 CF patients from 19 different countries demonstrated how the majority of pwCF may experience milder outcomes from SARS-CoV-2 infection, despite the much larger number of hospitalized patients who underwent organ transplantation than patients who did not [12].

Confirming these retrospective observations, a recent study found that SARS-CoV-2 replication is reduced in cystic fibrosis transmembrane conductance regulator (CFTR)-mutated bronchial cells with respect to functional CFTR bronchial epithelial cells [13]. These observations indicate an important role of CFTR protein in the regulation of SARS-CoV-2 replication; thus, CFTR was suggested as a potential novel molecular target for innovative antiviral treatment.

In this study, we evaluated the effect of short-term (up to 48 hpi) incubation with the specific pharmacological inhibitors of CFTR (IOWH-032 and PPQ-102) in bronchial epithelial cells expressing native CFTR.

## 2. Materials and Methods

### 2.1. Cells, Virus, and Treatments

The CFBE41o- WT cell line used herein is a subclone obtained from transfection of the parental CFBE41o- cell line with an HIV-based lentiviral vector containing the wild-type CFTR gene. The parental CFBE41o- is a CF human bronchial epithelial cell line, derived from a CF patient homozygous for the ΔF508 CFTR mutation, which was immortalized using the SV40 plasmid with a defective origin of replication (pSVori-). To culture these cells, minimum essential medium (MEM, Gibco, Thermo Fischer, Monza, Italy) supplemented with 10% FBS (Euroclone, Milan, Italy), 1% Glutamine (GlutaMAX, Gibco, Thermo Fischer), and the appropriate amount of selective factor (Puromycin) was used.

MucilAir™ (Epithelix Sàrl, Geneva, Switzerland) is an in vitro cell model of the human airway epithelium cultured at the air–liquid interface and reconstituted using human primary cells at low passage (P1). It was kept in MucilAir^TM^ culture medium (Epithelix Sàrl) at the air–liquid interface (ALI) for 1 week after receipt. We used this fully differentiated bronchial epithelial 3D model from primary human cells derived from healthy donors’ (wt/wt-CFTR MucilAir™) (*n* = 3; age mean 46 ± 16) and cystic fibrosis patients’ homozygotes for ΔF508 (F508del/F508del-CFTR MucilAir™) (*n* = 3; age mean 28 ± 10).

Cells were infected with two different SARS-CoV-2 strains: SARS-CoV-2 B.1 strain (hCoV-19/Italy/BO-VB12/2020|EPI_ISL_16978127), replicated in Vero E6 cells as described by Ogando and colleagues [14], and SARS-CoV-2 BA.5.1 (Omicron) strain, expanded in Calu-3 cells.

Cells were treated with IOWH-032 (MedChemExpress LLC., Monmouth Junction, NJ, USA), a synthetic small hydrazide molecule designed to selectively inhibit the CFTR channel with a reported IC_50_ of 8 μM, acting at the external surface of CFTR [15]. This molecule, with antisecretory activity, was first studied to treat diarrhea; it entered Phase II clinical trials in 2013, but did not progress further in clinical development, despite being proven to be safe and well-tolerated in healthy volunteers [16].

To confirm that the antiviral effects were due to CFTR inhibition, PPQ-102 (Selleck Chemicals LLC, Houston, TX, USA), a well-known CFTR inhibitor with an IC_50_ of 90 nM, was tested [17]. This drug targets the intracellular nucleotide binding domain of CFTR and inhibits the CFTR-mediated chloride current in a reversible and voltage-independent manner. It showed efficacy in a mouse model of ADPKD [18].

Both IOWH-032 and PPQ-102 were initially prepared as 10 mM stocks diluted in DMSO.

### 2.2. Cytotoxicity Assay

In vitro cell viability was assessed after IOWH-032 and PPQ-102 treatment on CFBE WT cells. The assay was carried out according to the CellTiter 96^®^ AQ_ueous_ One Solution Cell Proliferation Assay (Promega, Madison, WI, USA) at 1 h, 24 h, and 48 h post-treatment.

Briefly, 10^4^ cells were seeded in 96-well plates and incubated at 37 °C, humidified, and 5% CO_2_. After 24 h of incubation, the different dilutions of IOWH-032 (0.1 μM, 1 μM, 5 μM, 7 μM, 10 μM, 20 μM, 30 μM, 50 μM, and 100 μM) and PPQ-102 (0.1 μM, 1 μM, 5 μM, 10 μM, 20 μM, 30 μM, 50 μM, and 100 μM) were added for a final volume of 100 μL. In parallel, we also considered cells treated with pure DMSO at the same concentrations as the two compounds. Cells were then incubated at 37 °C, humidified, and 5% CO_2_, and after 1 h, 24 h, and 48 h post-treatment incubation, 20 μL of CellTiter 96^®^ AQ_ueous_ One Solution Reagent was added to each well. After 2 h of incubation, absorbance was recorded at 490 nm using a 96-well plate reader. DMSO was used as a blank. The 50% cytotoxic concentration (CC_50_) was determined.

### 2.3. TEER Measurement

In order to assess the integrity of the tissue monolayer, transepithelial electrical resistance (TEER) variations were measured, both in the wt/wt-CFTR MucilAir™ cells and wt/wt-CFTR MucilAir™ cells treated with 10 μM IOWH-032, using a volt-ohm meter (EVOM2, Epithelial Volt/Ohm Meter for TEER) and STX 2 electrodes (World Precision Instruments, Sarasota, FL, USA), according the manufacturer’s instructions. Briefly, on the apical side of the insert, 200 μL of pre-warmed MucilAir^TM^ culture media was applied. The electrode was cleaned in 70% ethanol and equilibrated in saline solution (0.9% NaCl; 1.25 mM CaCl_2_; 10 mM HEPES) until the volt-ohm meter registered 0.00. The electrode was inserted, with the long stem entering through the gap of the insert and leaning on the bottom of the well, and the short stem was immersed in the culture media, above the apical surface, for measurement. Following the measurement, the medium was immediately removed from the apical side. The values were calculated and represented as Ω·cm^2^, based on the surface area of the inserts (0.33 cm^2^).

### 2.4. Antiviral Activity Evaluation

T25 cell flasks were prepared with CFBE WT cells and, an hour before infection, increasing concentrations of IOWH-032 (0.1 μM, 1 μM, 5 μM, 7 μM, 10 μM, 20 μM, and 30 μM) and PPQ-102 (0.1 μM, 1 μM, 5 μM, 10 μM, 20 μM, and 30 μM) were added. An untreated flask, an uninfected flask, and a DMSO-treated flask were also prepared. After 1 h of incubation at 37 °C and 5% CO_2_, the SARS-CoV-2 B.1 strain was inoculated into the cells at a multiplicity of infection (MOI) of 1, and the cells were incubated for 1 h at 37 °C and 5% CO_2_. After incubation, the media was changed with fresh medium with the restored IOWH-032 concentrations. The supernatant and cell samples were collected at 48 h post-infection (hpi).

For MucilAir^TM^ infection experiments, the apical sides were gently washed twice with pre-warmed OptiMEM medium (GIBCO, Thermo Fisher Scientific, Monza, Italy); then, inserts were treated, adding 10 μM IOWH-032 on wt/wt-CFTR MucilAir™, and incubated for 1 h at 37 °C and 5% CO_2_ before infection. Next, SARS-CoV-2, diluted in OptiMEM at MOI 1, was inoculated in both wt/wt-CFTR MucilAir™ and F508del/F508del-CFTR MucilAir™ and incubated for 1 h at 37 °C and 5% CO_2_. After incubation, the medium on the basolateral side was changed to a fresh medium containing IOWH-032 10 μM, and the inoculum from the apical side was removed to restore the ALI state. At 48 hpi, cells were harvested in lysis buffer (Promega, Madison, WI, USA) while supernatants from apical washes or basolateral media were collected.

The supernatant’s SARS-CoV-2 load was detected with the multiplex real-time PCR Allplex 2019-nCoV assay kit targeting the E, RdRp/S and N genes (Seegene, Seoul, Republic of Korea) following the manufacturer’s instructions, while RNA was extracted from cells with ReliaPrep™ RNA Miniprep Systems (Promega), retrotranscribed into cDNA with iScript™ Reverse Transcription Supermix for RT-qPCR (Bio-rad, Hercules, CA, USA), and then analyzed by real-time qPCR on a CFX96 Real-Time System (Bio-Rad) using the primers indicated in Table 1.

The percentage of inhibition of SARS-CoV-2 replication was estimated for each drug concentration, and the half-maximal inhibitory concentration (IC_50_) was determined.

To verify that CFTR inhibitors’ antiviral activity is not SARS-CoV-2 strain-specific, the supernatant analyses described above were performed on treated WT CFBE41o^−^ cells infected with the BA.5.1 strain of SARS-CoV-2.

### 2.5. Inhibition Stage Determination

In addition to the “full-time” treatment (described above), two different treatment protocols were employed in order to shed light on the replication cycle phase targeted by CFTR inhibitors to inhibit viral replication. The first treatment was based on treatment performed with either 10 µM IOWH-032 or 20 µM PPQ-102 on WT CFBE41o^−^ for 1 h before viral infection and maintained during the viral incubation. The medium was changed after the 1 h of viral incubation with a fresh medium without a CFTR inhibitor. The second treatment was carried out with either 10 µM IOWH-032 or 20 µM PPQ-102 added after the 1 h of viral incubation, and then maintained until the collection of the supernatant at 48 hpi. The supernatant’s viral load was determined as previously described.

### 2.6. Electrophysiological Measurements

CFBE41o- WT cells were seeded on Costar Transwell^®^ inserts (Corning, NY, USA) and cultured in the air-liquid interphase (ALI) for two weeks. Once ready, the inserts were mounted on a slider in an Ussing chamber, and the transepithelial short-circuit current (Isc) was monitored using an EVC4000 multi-channel voltage/current clamp (WPI, World Precision Instruments). The two half chambers were filled with Quinton saline buffer solution. Subsequently, the ENaC blocker Amiloride (10 µM) was added to the apical side, and the cAMP analog CPT (100 µM), as well as the CFTR inhibitors IOWH-032 (10 µM) and PPQ-102 (10 µM), were added to both the apical and the basolateral sides. The tracings were recorded with PowerLab (8/35, AD Instruments, Sydney, Australia), and data analyses were performed using Labchart v8 software (AD Instruments).

## 3. Results

### 3.1. Cytotoxicity Evaluation

In order to exclude the possibility that the antiviral effects recorded in this experimental model may be related to cell toxicity, we evaluated the cell cytotoxicity of different concentrations of IOWH-032 and PPQ-102 in the CFBE41o- WT cell line (Figure 1).

We assessed several IOWH-032 concentrations (0.1, 1, 5, 7, 10, 20, 30, 50, and 100 μM) at different time points (1, 24, and 48 h post-treatment) and evaluated cell cytotoxicity with the CellTiter 96^®^ AQ_ueous_ One Solution Cell Proliferation Assay (Promega, Madison, WI). The analysis of cytotoxicity reported a CC_50_ > 50 μM at 48 h post-treatment (Figure 1a). No cytotoxicity was detected using the vehicle (DMSO) at the same concentration used for the preparation of the tested drugs. No significant cytotoxicity of IOWH-032 was reported at the previously reported CFTR inhibitory concentration IC_50_ [15]. Similar results were obtained for PPQ-102, with no significant cytotoxicity detected below 20 μM (Figure 1b).

We then selected the 10 µM concentration and tested the effect of the treatment with IOWH-032 on wt/wt-CFTR MucilAir^TM^, a fully differentiated 3D model bronchial epithelium. Given the limited number of 3D models available, we assessed tissue integrity by measuring transepithelial electrical resistance (TEER) at 0, 24, and 48 h after treatment to confirm the lack of cytotoxicity which was also seen in this experimental model.

TEER response after the IOWH-032 treatment of wt/wt-CFTR MucilAir™ (Figure 2), expressed as a percentage of the control, demonstrates that there is no significant reduction in tissue integrity after treatment.

### 3.2. Anti-SARS-CoV-2 Activity of CFTR Inhibitors

To assess the antiviral activity of CFTR inhibitors IOWH-032 and PPQ-102, we firstly analyzed SARS-CoV-2 B.1 RNA content using quantitative real-time RT-PCR, targeting the E viral gene at 48 hpi. We recorded a concentration-dependent viral inhibition in the cell culture supernatant by IOWH-032 (Figure 3a), reaching nearly 100% viral inhibition at 30 μM, with an IC_50_ of 4.52 μM (95% CI 3.24–5.71) (Figure 3b).

To determine which step of the viral life cycle is inhibited by the treatment, we also tested the intracellular SARS-CoV-2 content (Figure 3c). IOWH-032 treatment caused a viral inhibition of nearly 100% at 30 μM with an IC_50_ of 6.66 μM (95% CI 5.72–8.67), slightly higher than the concentration causing 50% inhibition in the supernatant fraction (Figure 3d).

PPQ-102 antiviral activity was also detected in the supernatant following a concentration-dependent increase in the percentage of inhibition, approaching 100% at 30 μM (Figure 4a), and an IC_50_ of 15.92 μM (95% CI 11.99–45.87) (Figure 4b).

At the intracellular level, the inhibition trend is still concentration-dependent, with an IC_50_ of 12.1 μM (95% CI 4.48–14.01) (Figure 4d), even though viral inhibition does not reach a value close to 100% at 30 μM (Figure 4c).

As no significant cell cytotoxicity was recorded at drug concentrations around 10 μM (Figure 5), we can conclude that the antiviral effect is not related to the cytotoxic effect that occurs at higher concentrations.

We then tested the antiviral activity of IOWH-032 at a concentration of 10 μM on wt/wt-CFTR MucilAir™, analyzing both the supernatant (Figure 6a) and intracellular (Figure 6c) SARS-CoV-2 RNA content. F508del-CFTR cells were used as a reference control of CFTR inhibition.

Even for this cellular model, we detected strong viral inhibition in the supernatant (Figure 6b), while at the intracellular level, the viral inhibition, although greater than 50%, was less consistent (Figure 6d). It is important to note that the viral inhibition associated with treatment in the supernatant is comparable to the effect of a dysfunctional CFTR (F508del-CFTR MucilAir™ cells, Figure 6b). On the other hand, there is a significant difference in viral inhibition at the intracellular level between these two cell types (Figure 6d).

It is noteworthy that the reduced viral titer induced by IOWH-032 in both the WT CFBE41o^−^ and wt/wt-CFTR MucilAir^TM^ cells was significantly more consistent in the supernatant than intracellularly, suggesting that CFTR inhibition can negatively affect the final steps of the viral cycle and the viral release from infected cells (Figure 7).

### 3.3. CFTR Inhibitors Affect Post-Entry Stages of SARS-CoV-2 Infection

To assess the antiviral activity of CFTR inhibitors and viral replication stages, we treated the cell cultures with two different protocols (Figure 8). In the first protocol, we infected the cell culture in the presence of either IOWH-032 or PPQ-102 for 1 h only to investigate the consequence of CFTR inhibition in the first phase of the viral replication cycle. It is noteworthy that viral inhibition was very low. In the second protocol, IOWH-032 or PPQ-102 were added to the cells one hour after exposure to the virus. In this case, both CFTR inhibitors significantly impacted the viral replication.

### 3.4. CFTR Inhibitors’ Antiviral Activity Is Not SARS-CoV-2 Strain-Dependent

To verify that the antiviral activity of CFTR inhibitors is not SARS-CoV-2 strain-specific, we repeated the supernatant analysis on treated WT CFBE41o^−^ cells infected with the SARS-CoV-2 BA.5.1 strain (Figure 9). It is noteworthy that CFTR inhibitors’ antiviral activity is still important, even against the SARS-CoV-2 Omicron variant.

The additional infection protocols confirm the data obtained with SARS-CoV-2 B.1 (Figure 10).

### 3.5. Electrophysiological Measurements

To confirm the capability of the CFTR inhibitors to efficiently inhibit CFTR channel function, in Ussing chambers, we assessed the transepithelial ion transport in the same setting, i.e., a polarized monolayer derived from CFBE41o- WT cells (Figure 11). Firstly, the epithelial sodium channel (ENaC) was transiently blocked by amiloride in order to measure only anion transport. The following addition of CFTR inhibitors IOWH-032 and PPQ-102 at a concentration of 10 μM induced a block in anion secretion through the monolayers, causing a current lowering (32.9 ± 14 and 48.8 ± 14 μA, respectively). To assess the residual CFTR functionality after the inhibition, a CFTR activator, CPT (100 μM), was added to a control monolayer (without adding the inhibitor) and to both the IOWH-032- and PPQ-102-treated monolayers. No anion secretory response was elicited by the PPQ-102-treated monolayers, confirming a complete blockade of the CFTR channel. While the cAMP agonist induces an increase in CFTR function in untreated cells, only a limited signal was detectable in IOWH-032-treated monolayers (16.6 ± 4 and 7.8 ± 5 μA, respectively, *n* = 3). These comparative results suggest a lower CFTR-inhibitory capacity of IOWH-032 compared to PPQ-102, supporting the limited efficacy of this inhibitor when used in vivo for the treatment of cholera [16].

## 4. Discussion

Several studies reported a significantly lower severity, number of infection cases, and viral spread of SARS-CoV-2 in pwCF than in the normal population, even though pwCF are expected to be at an increased risk of developing severe manifestations of COVID-19, since they are considered “fragile” patients [19,20,21,22,23]. Several factors can be associated with lower SARS-CoV-2 infection levels, such as preventive measures with which pwCF are familiar and the use of CF therapies. However, a recent study showed that SARS-CoV-2 infection is less productive in cells with CFTR defects than in healthy ones, suggesting that CFTR dysfunction may play a direct or indirect role in the viral replication cycle or its maturation [13].

It is known that chloride channels are involved in several steps of the viral life cycle, such as viral entry, membrane fusion processes, endosomal trafficking, and viral replication [24]. The CFTR channel has been reported to play a role in several respiratory viral infections; some studies [25,26] have shown that the influenza virus M2 matrix protein decreased CFTR activity and expression in bronchial epithelial cells. Panou et al. [27] reported a reduced Polyomavirus BK (BKPyV) infection associated with the inhibition of CFTR function, with CFTR being required during BKPyV transport to the endoplasmic reticulum.

Interestingly, TMEM16A and TMEM16F, calcium-activated Cl^−^ channels, are involved in viral replication; the inhibition of TMEM16A leads to reduced respiratory syncytial virus (RSV) gene expression [28] and can be activated during SARS-CoV-2 infection. Their inhibition with niclosamide causes the inhibition of the SARS-CoV-2 spike protein-driven fusion of the syncytia [29].

Epidemiological and early mechanistic studies revealed a role for CFTR in SARS-CoV-2 replication/function. As CFTR-defective cells have a complex array of dysfunctions associated with altered CFTR processing and a lack of activity, we focused on the short-term chemical inhibition of CFTR function in native cells to evaluate whether an altered CFTR function, more than alteration of additional pathways present in chronically CFTR defective cells, was involved. We chose two highly selective CFTR inhibitors which have been widely used in the literature, namely IOWH-032 and PPQ-102, to evaluate their antiviral activity.

We found that the antiviral activity of IOWH-032 and PPQ-102 occurred at IC_50_ values of 4.52 μM and 15.92 μM, respectively, which are lower than those causing significant cytotoxicity. Importantly, CFTR inhibitor antiviral activity was also confirmed against two viral strains, including the Omicron SARS-CoV-2 variant.

The disruption of the SARS-CoV-2 replication cycle and alteration in protein structure and function may be caused by a shift in ionic balance and intracellular pH variation.

According to a recent study, the endolysosome proteases TMPRSS-2 and cathepsins B and L activate the SARS-CoV-2 S protein in an acidic environment, which is why the deacidification of this organelle has been found to inactivate proteases and prevent viral infection [30,31]. The CFTR protein is necessary to maintain an intracellular ionic and pH balance and, based on the results reported, its dysfunction might represent an important factor driving reduced SARS-CoV-2 replication [32,33].

In particular, SARS-CoV-2 controls autophagosomal machinery to facilitate the creation of double-membrane vesicles, effectively subverting autophagy to increase replication [34,35,36]. Interestingly, CFTR dysfunction leads to BECN1 inactivation, perturbing endosomal fusion/maturation and trafficking, with a negative impact on intracellular trafficking. This leads to defective autophagosome formation [37,38,39,40], preventing SARS-CoV-2 from being able to take advantage of this pathway. In a study by Merigo et al., it was found that SARS-CoV-2 infection in malfunctioning CFTR cells caused the onset of autophagosomal structures containing cellular recycling material rather than replicative structures, as it did in wild-type cells [41].

As we have just observed, many pathways could be involved in reduced SARS-CoV-2 replication. In an attempt to determine which stage of the SARS-CoV-2 life cycle is involved in this reduced replication, we assessed the impact of treatments at different time-points. According to our results, CFTR inhibitors did not significantly impact SARS-CoV-2 cell entry; however, they were significantly more effective in the post-entry phase. In addition, comparing the intracellular and supernatant viral loads after CFTR inhibition, we found a stronger inhibitory effect in the supernatant, thus suggesting a major SARS-CoV-2 suppression effect during the intracellular phases of the SARS-CoV-2 replication cycle.

SARS-CoV-2 virions can be released from an infected cell via the Golgi compartment [42,43], the function and structure of which seem to be altered to facilitate viral trafficking [44], or by incorporation into deacidified lysosomes [45]; both are pathways that can be adversely impacted by pH deregulation, as in the case with CFTR alteration [46,47]. An ultrastructural study revealed that small secretory vesicles carrying a single virus particle are the main mechanism of SARS-CoV-2 egress [48]. Recent findings have demonstrated that SARS-CoV-2 exploits extracellular vesicles (EVs) for cellular exit and intercellular communications [49]; however, in this scenario, environmental pH stress and altered lipogenesis, possibly caused by CFTR disfunction, can cause a decrease in the number of EVs [50].

Interestingly, treatment with IOWH-032 reaches an almost 100% inhibition of SARS-CoV-2 mRNA levels, but at a concentration that displays significant cytotoxicity. Hence, whether an incomplete limitation of SARS-CoV-2 replication is enough to determine a significant clinical impact remains to be determined.

In conclusion, in this study, we demonstrated the significant anti-SARS-CoV-2 activity of CFTR inhibitors occurring during intracellular viral replication and below cytotoxic concentrations. Given that both CFTR inhibitors demonstrated strong and similar antiviral activity in two models of native bronchial epithelial cells and against two different SARS-CoV-2 strains, we can safely conclude that the loss of CFTR function may be considered an important factor in SARS-CoV-2 replication, suggesting a role for CFTR in SARS-CoV-2 infection. Although further studies are necessary to understand whether the molecular pathways governing CFTR-dependent SARS-CoV-2 replication may be considered as a clinically relevant target for antiviral treatment, this study supports the repurposing of CFTR inhibitors from diarrheal disease to antiviral application.

## Figures and Tables

**Figure 1 cells-12-00776-f001:**
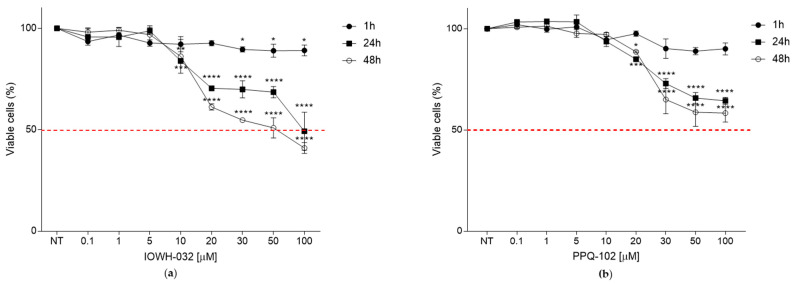
Toxicity evaluation of IOWH-032 and PPQ-102 (0.1, 1, 5, 10, 20, 30, 50, and 100 μM) at 1, 24, and 48 h post-treatment, evaluated on WT CFBE41o^−^ cells. (**a**) Graphical representation of IOWH-032 concentration-dependent toxicity, represented as a percentage of viable cells. The red dashed line represents the CC_50_; (**b**) graphical representation of PPQ-102 concentration-dependent toxicity as a percentage of viable cells. The red dashed line represents the CC_50._ (*n* = 3, * *p* < 0.05, ** *p* < 0.01, *** *p* < 0.001, **** *p* < 0.0001). Data are presented as the mean ± SD from three independent experiments, each of which was performed in triplicate.

**Figure 2 cells-12-00776-f002:**
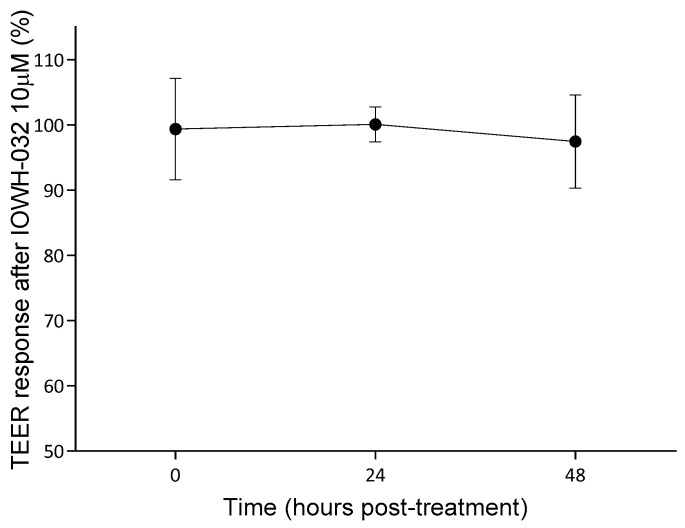
TEER response, presented as a percentage of the control, of wt/wt-CFTR MucilAir™ after treatment with 10 μM IOWH-032, measured after 0, 24, and 48 h post-treatment. Data are shown as the mean and SD of three independent experiments (*n* = 3).

**Figure 3 cells-12-00776-f003:**
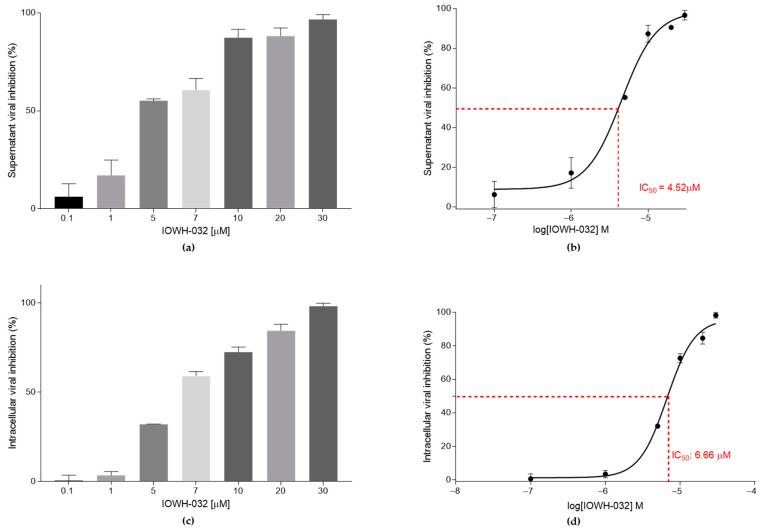
IOWH032 (0.1, 1, 5, 7, 10, 20, and 30 μM) anti-SARS-CoV-2 activity was studied in CFBE41o^™^ WT cells. The data are presented as the mean ± SD from three independent experiments. (**a**) Cellular supernatant’s SARS-CoV-2 inhibition at different IOWH-032 concentrations, presented as a percentage of the control CFBE41o^™^ WT untreated cells; (**b**) graphical representation of supernatant IOWH-032 IC_50_; (**c**) intracellular SARS-CoV-2 inhibition at different IOWH-032 concentrations, presented as a percentage of the control CFBE41o^™^ WT untreated cells; (**d**) graphical representation of the intracellular IOWH-032 IC_50_.

**Figure 4 cells-12-00776-f004:**
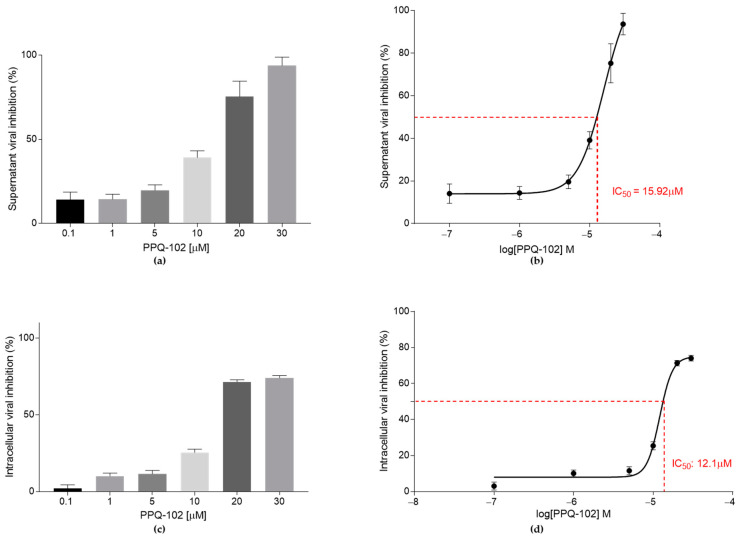
PPQ-102 (0.1, 1, 5, 10, 20, and 30 μM) anti-SARS-CoV-2 activity in CFBE41o^−^ WT. Data are presented as the mean ± SD from three independent experiments. (**a**) Supernatant’s SARS-CoV-2 inhibition at different PPQ-102 concentrations, presented as a percentage of the control CFBE41o^−^ WT untreated cells; (**b**) graphical representation of the supernatant IC_50_ determination of PPQ-102; (**c**) intracellular SARS-CoV-2 inhibition at different PPQ-102 concentrations, presented as a percentage of the control CFBE41o^−^ WT untreated cells; (**d**) graphical representation of the intracellular IC_50_ determination of PPQ-102.

**Figure 5 cells-12-00776-f005:**
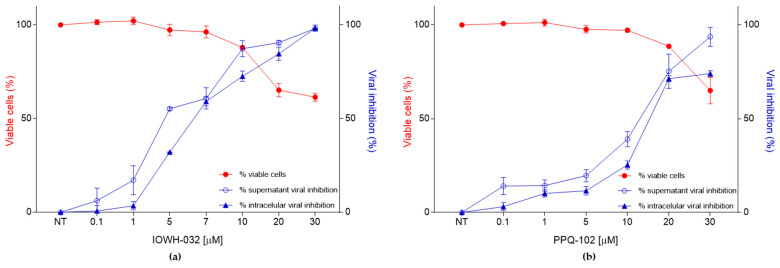
Combined data of WT CFBE41o^−^ cells’ viability and viral inhibition after CFTR inhibitor treatment 48 hpi. (**a**) Comparison between the percentage of viable cells (red line), percentage of supernatant viral inhibition (blue line, empty circles), and percentage of intracellular viral inhibition (black line, triangle) after IOWH-032 treatment; (**b**) comparison between the percentage of viable cells (red line), percentage of supernatant viral inhibition (blue line), and percentage of intracellular viral inhibition (black line) after PPQ-102 treatment.

**Figure 6 cells-12-00776-f006:**
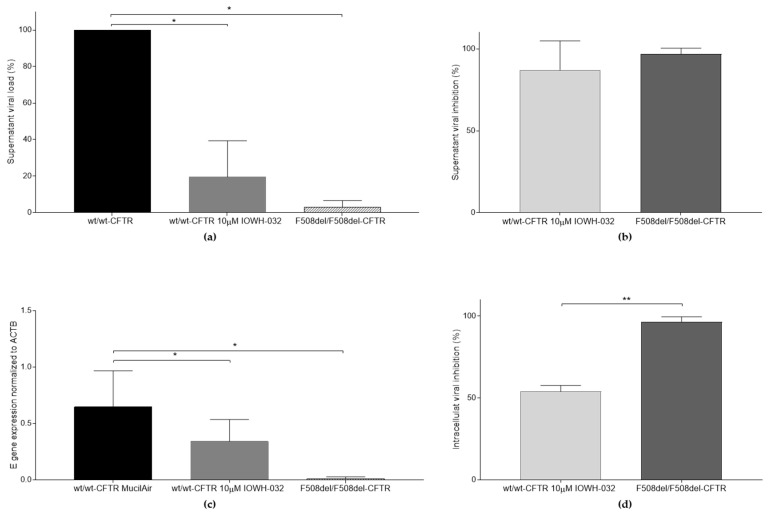
Comparison of SARS-CoV-2 replication between wt/wt-CFTR, wt/wt-CFTR with 10 μM IOWH-032, and F508del/F508del-CFTR MucilAir™. (**a**) Supernatant’s viral load of wt/wt-CFTR, wt/wt-CFTR treated with 10 μM IOWH-032, and F508del/F508del-CFTR MucilAir™, expressed as a percentage (* *p* < 0.05); (**b**) supernatant’s viral inhibition, presented as a percentage of the control untreated wt/wt-CFTR MucilAir™; (**c**) intracellular E gene expression, normalized to ACTB (β-Actin) of SARS-CoV-2 infected cells (* *p* < 0.05); (**d**) intracellular viral inhibition, presented as a percentage of the control untreated wt/wt-CFTR MucilAir™ (** *p* < 0.01). Data are presented as the mean ± SD from three independent experiments.

**Figure 7 cells-12-00776-f007:**
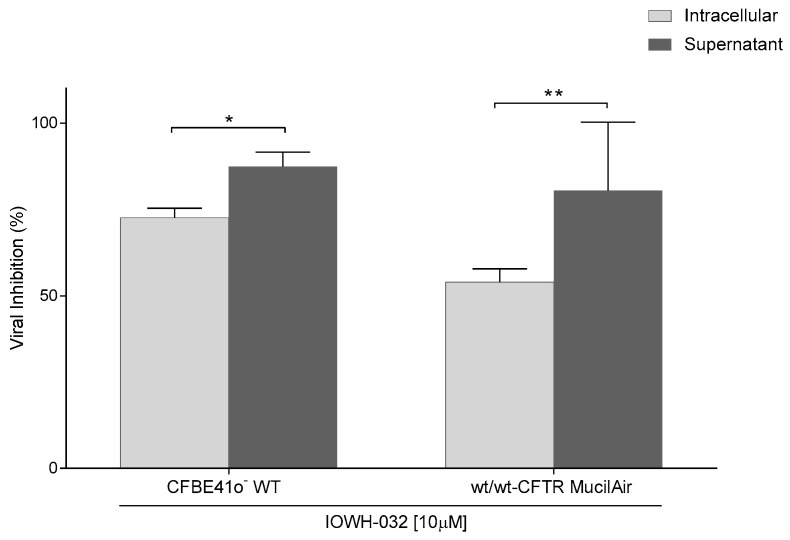
Comparison between intracellular and supernatant viral inhibition in both CFBE41o^−^ WT and wt/wt-CFTR MucilAir™ treated with IOWH-032 10 μM (*n* = 3, * *p* < 0.05, ** *p* < 0.01).

**Figure 8 cells-12-00776-f008:**
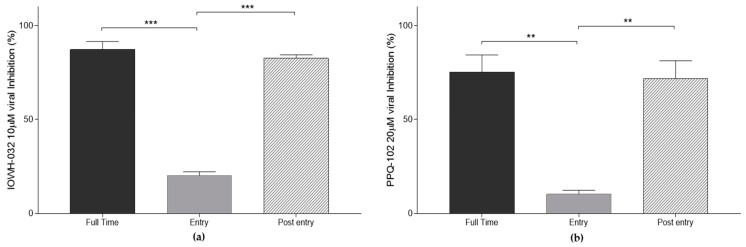
Antiviral activities of CFTR inhibitors after “full”, “entry”, and “post-entry” treatment in WT CFBE41o^−^ cells. Data are presented as the mean ± SD from three independent experiments. (**a**) IOWH-032 10 μM viral inhibition at different stages of SARS-CoV-2 infection; (**b**) PPQ-102 20 μM viral inhibition at different stages of SARS-CoV-2 infection. (*n* = 3, ** *p* < 0.01, *** *p* < 0.001).

**Figure 9 cells-12-00776-f009:**
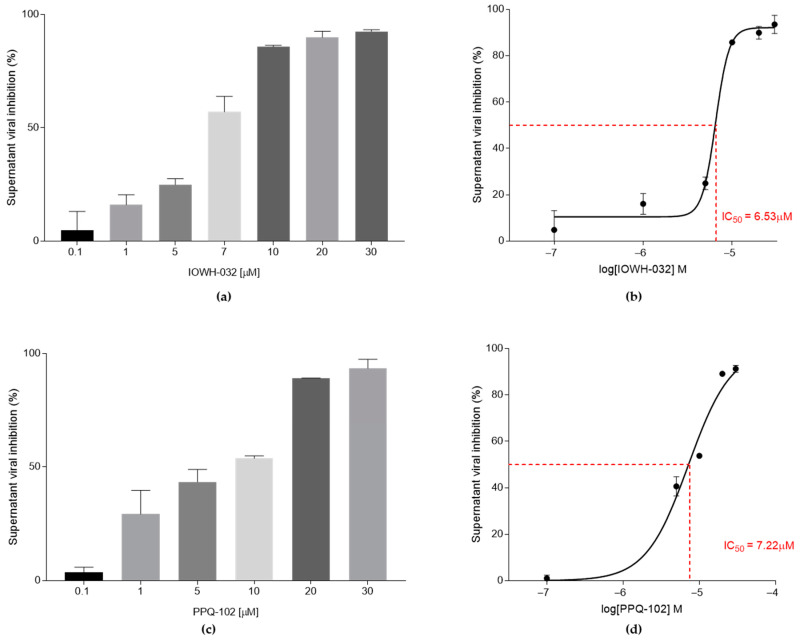
Antiviral activity of CFTR inhibitors against BA.5.1 variant on WT CFBE41o^−^. The data are presented as the mean ± SD from three independent experiments. (**a**) IOWH032 (0.1, 1, 5, 7, 10, 20, and 30 μM) anti-SARS-CoV-2 activity; (**b**) graphical representation of supernatant IOWH-032 IC_50_; (**c**) PPQ-102 (0.1, 1, 5, 10, 20, and 30 μM) anti-SARS-CoV-2 activity; (**d**) graphical representation of PPQ-102 IC_50_.

**Figure 10 cells-12-00776-f010:**
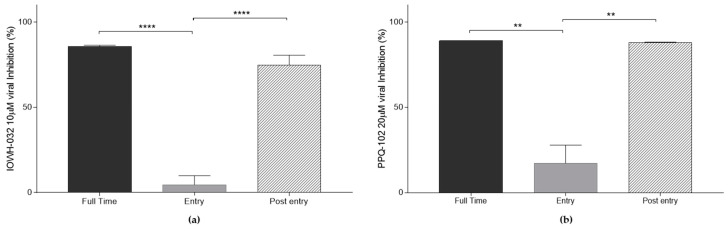
Antiviral activities of CFTR inhibitors against SARS-CoV-2 BA.5.1 after “full”, “entry”, and “post-entry” treatment in WT CFBE41o^−^ cells. Data are presented as the mean ± SD from three independent experiments. (**a**) IOWH-032 10 μM viral inhibition at different stages of SARS-CoV-2 infection; (**b**) PPQ-102 20 μM viral inhibition at different stages of SARS-CoV-2 infection. (*n* = 3, ** *p* < 0.01, **** *p* < 0.0001).

**Figure 11 cells-12-00776-f011:**
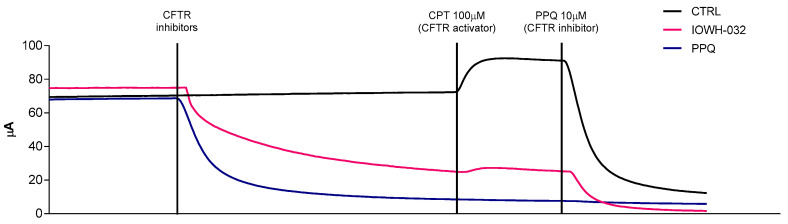
Short-circuit current (Isc) measurement tracing was used to prove the efficacy of IOWH-032 and PPQ-102 in inhibiting the CFTR channel in WT CFBE41o^−^. ENaC blocker amiloride (10 µM) was added on the apical side; the cAMP analog CPT (100 µM) and the CFTR inhibitors IOWH-032 (10 µM) and PPQ-102 (10 µM) were added to both the apical and the basolateral sides. A representative experiment of the three performed experiments is shown.

**Table 1 cells-12-00776-t001:** Primers used for intracellular RNA RT-qPCR.

Primers	Sequences (5′→3′)
E gene F	ACAGGTACGTTAATAGTTAATAGCGT
E gene R	ATATTGCAGCAGTACGCACACA
GAPDH F	TCAAGAAGGTGGTGAAGCAGG
GAPDH R	CAGCGTCAAAGGTGGAGGAGT
β-Actin F	CCCTGGACTTCGAGCAAGAG
β-Actin R	ACTCCATGCCCAGGAAGGAA

## Data Availability

Not applicable.

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
