# Peer review of "CFTR Inhibitors Display In Vitro Antiviral Activity against SARS-CoV-2"

_cells, 2023, doi:10.3390/cells12050776_

Round 1

Reviewer 1 Report

In the manuscript, the authors investigated the role of two CFTR inhibitors on the SARS-CoV-2 replication process and they concluded that two inhibitors significantly inhibit SARS-CoV-2 replication at the concentration without displaying cytotoxicity.  Overall, this is a well-written manuscript, and CFTR inhibitors display the potential to treat COVID-19, which will be of interest to the biochemists in the SARS-CoV-2 field. The conclusions from this manuscript are consistent with their previous work “CFTR Modulation Reduces SARS-CoV-2 Infection in Human Bronchial Epithelial Cells”. However, the mechanisms are lacking. The following questions should be addressed before acceptance to publish in the cells.

Major points:

1.    The novelty of the manuscript is limited due to their previous publication (https://doi.org/10.3390/cells11081347).

2.    There is still no good explanation for how CFTR inhibition reduces SARS-CoV-2 replication. In the Discussion section, the authors say that CFTR inhibitors may affect many steps. At least whether the inhibitors affect the viral entry or not should be answered by experiments. A pseudo-Spike viral entry assay is well established and should be done.

3.  The difference between intracellular inhibition and supernatant inhibition is not very dramatic. So, it will be good if a different method is utilized to compare the viral infection rate with/without inhibitor treatment to support the authors’ conclusion, for example, immunofluorescence or WB.

4.  The references 42 and 43 in line 355 should include the early and more related research articles, like PMCID: PMC7590812; PMCID: PMC8393858; PMCID: PMC8923104.

Minor points:

1.    There should be a space between number and unit, for example, line 20, 4.52uM should be 4.52 uM.

2.    In lines 79 and 80, the sentence is Grammarly incorrect.

Reviewer 2 Report

Lagni et al., reported that CFTR inhibitors showed antiviral effect on the infection of SARS-CoV-2 in restored cell line of human bronchial epithelial cell line, CFBE41lo-. The contents are interesting. But this article seems a continued work of the previous study from same group (Cell. 2022. 11(8)), however, the differences from it are unclear. I recommend the reconstruction of the article while differentiating from the previous study. The methodology of the study is also suspicious.

Major comments:

L90: The sequence ID (GISAID or NCBI) of used virus should be shown. It seems the used SARS-CoV-2 is the isolate of alpha variant. Currently, Wuhan type of viruses are exterminated naturally. The data with the omicron variant should be added so as to work the data for current SARS-CoV-2 situation. In addition, the virus stock was propagated using VeroE6 cells. Various study showed that the viral sequence easily mutated (especially in spike region) in the absence of serine protease (see 10.1371/journal.ppat.1009233, etc). Possibly, the data with VeroE6-propagated virus does not reflect the natural SARS-CoV-2 infection. It requires to reconsider the virus propagation method.

L133: The inhibitors were used as only “pretreatment”. In discussion section, the author implied these affect on virus replication, but only “pretreatment” never cleared the effect on virus replication. To clear what step is affected by the inhibitor, the time course of drug treatment should be considered carefully. Real-time RT-PCR sets for the detection of subgenomic mRNA should be used for the evaluation of viral replication. If the cells were pretreated with the drug (and virus infection with drug), and cells were cultured without inhibitor, it will reveal the effect on virus entry. If the cells were treated with the drug after virus infection, it will reveal the effect on virus replication.

Figure 1: There is no data for CFBE41o- WT cells.

Figure 2: There is no data for F508del-CFTR MucilAir.

Figure 5: Are the graph combined the data of the viability of CFBE41o- cells and the viral inhibition with CFBE41o- WT cells?

Minor comments:

L30: The “coronavirus disease 2019” is correct for “COVID-19”. The “coronavirus illness 2019” will become “COVII-19”.

L51: Molnupiravir is a nucleoside analogue, and not a polymerase inhibitor.  

L66: It requires to be defined that the “CFTR” stand for (Cystic Fibrosis Transmembrane conductance Regulator?).

L83: It seems the MucilAir is consist of a single layer of differentiated epithelial cells. There are no basal fibroblasts, blood vessel epithelium in lower layer. What is the “3D”?

L152: It seems the Allplex 2019-nCoV assay kit applies three Corman’s assays. Why is the probe not listed in Table 1? If only primers were used in SYBR green assay, the methods should be stated correctly.

Figure 4 legend: anti SARS-CoV-2 activity in CFBE”41o-“ WT?

Round 2

Reviewer 1 Report

The revised manuscript is greatly improved. The only shortcoming is the novelty due to the authors' previous publication. I am fine if the manuscript is accepted for publication after a grammar and spelling check. A minor error is that in line 190, The media was should be The media were.

Reviewer 2 Report

I agree to the modifications.